# Polymorphisms of *TXK* and *PLCE1* Genes and Their Correlation Analysis with Growth Traits in Ashidan Yaks

**DOI:** 10.3390/ani14233506

**Published:** 2024-12-04

**Authors:** Juanxiang Zhang, Xita Zha, Guowu Yang, Xiaoming Ma, Yongfu La, Xiaoyun Wu, Xian Guo, Min Chu, Pengjia Bao, Ping Yan, Chunnian Liang

**Affiliations:** 1Lanzhou Institute of Husbandry and Pharmaceutical Sciences, Chinese Academy of Agricultural Sciences, Key Laboratory of Yak Breeding Engineering of Gansu Province, Lanzhou 730050, China; 15103990593@163.com (J.Z.); guowu202302@163.com (G.Y.); maxiaoming@caas.cn (X.M.); layongfu@caas.cn (Y.L.); wuxiaoyun@caas.cn (X.W.); guoxian@caas.cn (X.G.); chumin@caas.cn (M.C.); baopengjia@caas.cn (P.B.); pingyanlz@163.com (P.Y.); 2Key Laboratory of Animal Genetics and Breeding on Tibetan Plateau, Ministry of Agriculture and Rural Affairs, Lanzhou 730050, China; 3Qinghai Qilian County Animal Husbandry and Veterinary Workstation, Qilian 810400, China; zhaxita@163.com

**Keywords:** *TXK* gene, *PLCE1* gene, growth traits, correlation analysis, Ashidan yak

## Abstract

Yaks are indispensable economic animals in pastoral areas, providing local herdsmen with abundant necessities of life. Their meat is of excellent quality and highly favored by the market. As a newly cultivated breed, the Ashidan yak has been under continuous selection and breeding to improve its production performance, which is a long-term goal. The growth traits of yaks are crucial indicators of their economic benefits. Therefore, through the use of SNP molecular marker-assisted selection technology, the early and precise selection of yak growth traits can be achieved. This technology effectively accelerates the pace of genetic improvement and enhances breeding efficiency.

## 1. Introduction

The yak (*Bos grunniens*) is a rare and precious genetic resource in China [1], capable of providing herdsmen with meat, milk, wool, fuel, and other necessities of life and production [2,3]. However, the development and utilization of yak resources in China are constrained by the harsh environment of the plateau, the severe degradation of grasslands, and the slow growth rate of yak breeds [4]. The Ashidan yak, a newly cultivated breed in China, demonstrates favorable growth traits in its offspring, with the improvement of these traits being intimately tied to advancements in modern biotechnology [5]. As an important means of molecular marker-assisted breeding, SNPs are playing an increasingly significant role in yak genetic breeding. In recent years, by analyzing SNPs in the yak genome, researchers have deduced the origins, differentiation, and migration patterns of yaks [6], as well as revealed the genetic structure, genetic variation, and genetic distances among yak populations [7]. In addition, these SNPs can be used for yak breed improvement and breeding for economic traits [8,9]. In animal breeding practice, growth traits are key indicators to measure breeding potential and economic value, which are directly related to the production efficiency and economic benefits of animal husbandry [10]. Since the combination of yak breeding and MAS (molecular marker-assisted selection) technology, researchers have successfully found several gene loci closely related to growth traits. In an association analysis of yak body weight, SNP sites such as AX-174555047 were confirmed as molecular breeding markers with potential application value [11]. Significant SNPs associated with yak growth traits were also identified in the exon regions of *FOXO1* and *FOXO3* genes [12]. Consequently, leveraging SNP marker-assisted selection technology to precisely breed for these traits is of paramount importance for enhancing the genetic quality and production performance of yak breeds.

Tyrosine protein kinase (*TXK*), a prominent member of the Tec kinase family [13], is characterized by a unique long amino-terminal region abundant in pleckstrin homology (PH) and Tec homology (TH) domains [14]. These specific domains are pivotal in facilitating the TCR/CD28 signaling-induced actin cytoskeleton reorganization process [15]. Recent advancements have illuminated the role of the *TXK* gene in the economic traits of livestock and poultry. Abril-Parreño and colleagues, through genome-wide association analysis (GWAS) in cattle, revealed a significant association between the *TXK* gene and semen quality traits in bulls [16]. In pig research, the *TXK* gene has been implicated in close correlation with blood traits [17]. Dadousis et al. successfully identified TXK as a key candidate gene affecting body weight at 35 days of age by performing a genome-wide association study in broilers [18,19]. Furthermore, studies conducted on Charolais beef cattle have demonstrated a notable impact of the *TXK* gene on body weight at one year of age [20].

Phospholipase C ε1 (*PLCE1*), as a key member of the phospholipase C (PLC) family, plays a central role in the regulation of cell growth, apoptosis, proliferation and differentiation [21]. When PLCE1 is activated, it will produce 1, 4, 5-triphosphate (IP3), which mediates the release of calcium through its specific receptor. This process indirectly promotes the development and regeneration of skeletal muscle and participates in the mechanism of muscle contraction [22]. Although most studies on PLCE1 have focused on cancer, its importance in animal growth and development is also receiving increasing attention. Lan et al. [23] found that the *PLCE1* gene is present in a pathway related to muscle growth and that the *PLCE1* gene helps to promote muscle fiber increase during pectoral muscle development in plateau turtle doves. A study on Indian cattle revealed an association between the *PLCE1* gene and production traits in these animals [24]. In another genome-wide association study on cattle, *PLCE1* was clearly identified as a candidate gene closely related to milk yield and reproductive traits [25].

In conclusion, the *TXK* and *PLCE1* genes are closely associated with economic traits in animals. However, there are still some gaps in the study of the correlation between these two genes and yak growth traits. The objective of this research was to explore the correlation between SNP loci in the *TXK* and *PLCE1* genes and growth characteristics of Ashidan yaks, ultimately contributing genetic resources for marker-assisted selection in yak breeding programs.

## 2. Materials and Methods

### 2.1. Experimental Animals and Traits

The experimental population selected in this study was Achidan yaks from Qinghai Provincial Yak Breeding and Extension Service Center (longitude 102.44°, latitude 37.92°). During the experiment, the yaks were fed with year-round grazing and supplemented with forage during the cold season to ensure that they had the same living environment and similar age. During the experimental period, 5 mL blood samples were collected from each yak, and these samples were stored in anticoagulant tubes at −20 °C. According to the national standard of the People’s Republic of China “Technical Specification for Determination of Yak Productivity” (GB/T 43842-2024 [26]), we measured the growth traits of 232 yaks at different growth stages (6, 12, 18, and 30 months of age). Body weight (BW), wither height (WH), body oblique length (BL), and chest girth (CG) were measured outdoors.

### 2.2. DNA Sample Extraction

The blood samples collected from 232 Ashidan yaks were thawed naturally at 4 °C. Simultaneously, a metal bath was preheated to 56 °C. The extraction of DNA was performed following the guidelines outlined in the Blood Genomic DNA Extraction Kit’s instructions (DP341, Tiangen Biochemical Technology Co., Ltd., Beijing, China). Following the completion of the extraction, the quality and concentration of the DNA samples were assessed using a NanoDrop instrument (Thermo, Beijing, China) to ensure that the DNA concentration exceeded 50 ng/μL and that the OD260/OD280 ratio fell within the acceptable range of 1.7 to 1.9.

### 2.3. Genetic Typing

We used cGPS (Genotyping by Pinpoint Sequencing of liquid captured targets) liquid microarray technology to detect the genotypes of 232 individual Ashidan yaks [27]. cGPS is a high- and medium-density targeted sequencing genotyping technology independently developed by the Huazhi Biological Company. An optimized thermodynamic stability algorithm model was utilized to design specificity probes targeted at the desired interval sequences. These probes were synthesized and subsequently utilized in liquid-phase hybridization for the capture and enrichment of multiple unique target sequences located at unoccupied genomic positions. Following this, second-generation sequencing techniques were applied to the enriched target intervals to comprehensively identify all SNP/INDEL site genotypes present. The raw Reads data underwent rigorous quality control using Fastp software (v0.23.4), which filtered out low-quality Reads to refine the dataset. Read pairs containing more than 50% of bases with a quality score of Q ≤ 20 were excluded, as well as Reads containing an excessive number of N bases (exceeding 5 Ns) or those with a length of less than 100 bases, thereby ensuring a high degree of data integrity. The genomic localization of the SNPs was accurately determined through alignment with the yak reference genome Bosgru v3.0 (GCA_005887515.1) [28].

### 2.4. SNPs Validation

In this study, the *TXK* (g.55,999,531C>T) and *PLCE1*(g.342,350T>G) gene loci were screened by whole gene association of cGPS data. The genotyping results of the above two gene loci were verified by Polymerase Chain Reaction (PCR) amplification. According to the sequences of yak *TXK* (accession number: ENSBGRG00000001037) and *PLCE1* (accession number: ENSBGRG00000024803) published in the Ensembl database, the PCR primers were crafted utilizing Primer Premier 5.0 software developed by Premier Biosoft International, located in San Francisco, CA, USA. The two loci were positioned on chromosomes 6 and 25 of the yak reference genome Bosgru v3.0, which has the accession number GCA_005887515.1.

Primer information is shown in Table 1, which was synthesized by Biological Company (Xi’an, China). The PCR reaction mixture consisted of 40 μL, comprising 20 μL of 2 × Super Pfx Master Mix, 1 µL of DNA template with a concentration of 100 ng/µL, 1 µL of each of the forward and reverse primers (both at 10 µmol/L), and 17 µL of sterile, enzyme-free water. The PCR procedure was as follows: pre-denaturation at 94 °C for 30 s; denaturation at 98 °C for 10 s, annealing at 57 °C for 15 s, extension at 72 °C for 2 min, 35 cycles; extension at 72 °C for 1 min; and storage at 4 °C. PCR products were detected on a 1% agarose gel, and those that matched the expected bands were sent to Qingke (Xi’an, China) for sequencing. The results were analyzed using MEGA 7.0 and BioEdit (7.2.5.0) software [29].

### 2.5. Statistical Analysis

Utilizing GDICALL online software (accessible at http://www.msrcall.com/Gdicall.aspx, last accessed 25 September 2024), we calculated various genetic parameters for the two loci, including heterozygosity (HE), the number of effective alleles (NE), polymorphism information content (PIC), genotype distribution, and allele frequency. We computed the *p*-values from both the chi-square test and the Hardy–Weinberg equilibrium test. We used an online tool (https://www.bioinformatics.com.cn/, last accessed on 14 September 2024) to produce a map of the genotype distribution of the SNP loci.

Using IBM SPSS Statistics 25 software (IBM, Armonk, NY, USA), we conducted a one-way ANOVA to investigate the relationship between polymorphisms in the *TXK* and *PLCE1* genes and growth traits in the yaks. To analyze the factors influencing yak production traits, we employed a simplified general linear model based on practical considerations. The simplified model is presented in Equation (1), where Yi represents the phenotypic value of a growth trait, µ is the overall mean of the growth trait, SNPi is the fixed effect of the genotype class at a specific locus, and e represents the random error effect. Pairwise comparisons between groups were performed using the Least Significant Difference (LSD) method, and the results are presented as mean ± standard deviation (mean ± SD). Statistical significance was considered at *p* < 0.05.
Yi = µ + SNPi + e(1)
where Yi = phenotypic value of growth traits, µ = population mean of growth traits, SNPi = fixed effect of the genotypic category of the locus, and e = random error effect.

## 3. Results

### 3.1. Genotyping Results and Genetic Parameters of TXK and PLCE1 Loci in Ashidan Yaks

The genotype frequencies, allele frequencies, and polymorphism information content (PIC) for the two SNP loci within the *TXK* and *PLCE1* genes are summarized in Table 2. The results indicate that these two SNP loci exhibit three distinct genotypes within the Achidan yak population; therefore, the genotyping results obtained from the cGPS chip were validated. The results are shown in Figure A1. The chip typing results were consistent with the sequencing results, indicating that the chip typing results were accurate. The distribution of genotypes of these two loci in the main production traits of the Ashidan yaks is shown in Figure 1. At the loci g.55,999,531C>T and g.342,350T>G, the distribution of the three genotypes is relatively even, suggesting that the frequencies of the genotypes are comparable and do not exhibit a significant skew towards any particular genotype.

At the g.55,999,531C>T locus, the genotype frequencies of CC, CT, and TT were 0.397, 0.474, and 0.129, respectively, and the genetic frequencies of alleles C and T were 0.634 and 0.366. The frequency content of the CT genotype was the highest, indicating that heterozygotes were dominant. He and Ne were 0.474 and 1.905, respectively, and PIC was 0.475, indicating that the locus had moderate polymorphism and was in Hardy–Weinberg equilibrium. At the g.342,350T>G locus of the *PLCE1* gene, there were three genotypes, TT, TG, and GG, with respective genotype frequencies of 0.487, 0.409, and 0.104. The allele frequencies for T and G were 0.692 and 0.308, respectively, indicating that T is the term major allele and TT is the term major genotype. The He and Ne values were 0.409 and 1.693, respectively. The *p*-value of 0.366 suggests that the locus was in Hardy–Weinberg equilibrium. The PIC of 0.426 indicates moderate polymorphism (0.25 < PIC < 0.5) at this locus.

### 3.2. Association Analysis of SNPs in TXK and PLCE1 Genes with Growth Traits in Ashidan Yaks

The results of the association analysis between the g.55,999,531C>T locus and growth traits of the Ashidan yaks are shown in Table 3. The results showed that this locus was significantly correlated with body length and chest circumference of the Ashidan yaks (*p* < 0.05). The body length of the CC genotype was significantly higher than that of the TT genotype at 6 months. The chest circumference of the 12-month-old CT genotype individuals was significantly higher than that of the TT genotype individuals.

The association analysis between the g.342,350T>G locus of the *PLCE1* gene and the growth traits of the Ashidan yaks is shown in Table 4. Specifically, individuals with the GG genotype exhibited significantly higher body height at 18 months of age compared to those with the TT and TG genotypes. However, no significant differences were observed in the growth traits of the remaining Ashidan yaks across different genotypes and at various growth stages.

## 4. Discussion

As an important economic trait of yaks, growth traits are directly related to meat yield, growth rate, and adaptability [30]. With the rapid advancement of modern molecular biology technology, SNPs, as third-generation molecular markers, have demonstrated significant potential for application in livestock and poultry genetic breeding. By screening for SNP markers that are closely associated with specific economic traits, breeders can select livestock and poultry at an early stage, thereby improving the efficiency of the selection process. In this study, we screened two SNP loci: one in the *TXK* gene (g.55,999,531C>T) and another in the *PLCE1* gene (g.342,350T>G). By analyzing the polymorphism information content of these two SNP loci, we discovered that the PIC values for g.55,999,531C>T and g.342,350T>G were 0.475 and 0.426, respectively, both of which fell within the range indicative of moderate polymorphism (0.25 < PIC < 0.5). This finding suggests that these two loci exhibit a high degree of genetic variability in Ashidan yak populations and can better reflect the genetic diversity of these populations [31]. Furthermore, the fact that these two SNP loci were in Hardy–Weinberg equilibrium indicates that the g.55,999,531C>T and g.342,350T>G loci have not been significantly impacted by mutation, selection, or genetic drift during the long-term evolutionary process. Consequently, their genotypes and gene frequencies have remained relatively stable within the population, demonstrating a certain degree of genetic superiority. The existence of this equilibrium state provides a solid foundation for genetic studies.

During the growth and development of yaks, body weight is the core index to measure their growth rate and overall health status. By accurately measuring body size parameters such as height, body length, and chest circumference, we can intuitively grasp the changes in body shape and development patterns of yaks. For the improvement of yak breeds, it is of great importance to mine key molecular markers closely related to growth traits. The *TXK* gene regulates the reorganization and remodeling of the actin cytoskeleton by responding to the TCR/CD28 signaling pathway and then profoundly affects the growth and development of tissues and organs. An association between the *TXK* gene and body weight at one year of age was demonstrated in a genome-wide association study (GWAS) of growth traits in Charolais beef cattle [20]. In addition, the *TXK* gene was also reported by previous studies as a candidate gene related to body weight in broiler chickens [18,19]. In this study, we further found that an SNP g.55,999,531C>T in the *TXK* gene was significantly associated with body length and chest circumference of Ashidan yaks. This finding is consistent with the results of previous studies and again confirms the close association between *TXK* gene polymorphisms and animal growth traits. However, it should be noted that in the present study, the *TXK* gene was mainly associated with the body size of yaks, but not directly with body weight. We speculate that this may be due to the differences in genetic background and physiological mechanisms among different animal breeds.

At present, there is no report on the role of the *PLCE1* gene in animal growth traits, but the present study found that the g.342,350T>G locus of the *PLCE1* gene was significantly associated with the body height of Ashidan yaks at 18 months of age. When the motif of this gene locus was mutated from T to G, mutant GG significantly improved the body height of Ashidan yaks at 18 months of age. Previous studies have shown that the *PLCE1* gene is associated with body weight [32]. Further studies have found that this gene can significantly affect the differentiation process of muscle cells and is related to muscle regeneration [33]. Although these studies were conducted in humans, the results of our study are similar, indicating that the *PLCE1* gene may also show similar timeliness in yak species and may indirectly affect animal growth and development, but the specific molecular mechanism needs to be further studied.

We found that both SNP sites of the *TXK* gene (g.55,999,531C>T) and the *PLCE1* gene (g.342,350T>G) are located in the intronic region of the genes. Despite this location, they did not affect their relationship with yak growth traits. Previous studies have found that there are three SNP sites in the intron region of Myogenic factor 5 (*MYF5*), which are closely related to the significant changes in chest circumference and body length of Qinchuan cattle [34]. At the same time, it has been reported that the growth traits of yaks are significantly affected by six SNP loci in the intron region of SMAD family member 4 (*SMAD4*) [35]. The above research is similar to the results of this study, which indicate that SNP loci exist with the intron region of the gene. Introns are non-coding regions in genes located between exons, and they play key roles in gene expression and regulation [18]. SNPs in introns may affect processes such as transcriptional regulation, splicing, and post-transcriptional modification of RNA [19]. They may lead to splicing variants that affect the way genes are spliced, resulting in different mRNA variants that affect protein structure and function. Therefore, the g.55,999,531C>T locus and the g.342,350T>G locus will have different effects on the growth traits of Ashidan yaks. The two SNP loci can serve as potential molecular markers for enhancing the growth traits of yaks.

We found that both SNP sites in the *TXK* gene (g.55,999,531C>T) and the *PLCE1* gene (g.342,350T>G) were located in the intronic region of the genes. There are three SNP sites in the intron region of Myogenic factor 5 (*MYF5*), which are closely related to the significant changes in chest circumference and body length of Qinchuan cattle [34]. It has been reported that the growth traits of yaks are significantly affected by six SNP loci in the intron region of SMAD family member 4 (*SMAD4*) [35]. This is consistent with the results of the present study, where SNP sites are present in the intronic region. These studies suggest that SNP sites in the intronic region are important for the regulation of growth traits in animals. Introns, as non-coding regions between gene exons, play a key role in gene expression and regulation [18]. Although they are not directly involved in protein coding, sequence changes in introns can affect processes such as transcriptional regulation, splicing, and post-transcriptional modification of RNA [19]. Therefore, these two SNP loci can be used as potential molecular markers for improving growth traits in yaks.

With the rapid development of molecular biology, MAS technology can accurately predict the genetic potential of individuals at an early stage by using genetic markers that are closely linked to the target trait, thus accelerating the breeding process. Based on this study, the SNP loci of the *TXK* and *PLCE1* genes can be used as potential molecular markers that are expected to bring changes to improved breeding efforts for growth traits in yaks. By using MAS technology, breeders can identify individuals with superior growth traits at the young stage of yaks and then conduct targeted breeding to improve the efficiency and accuracy of breeding.

## 5. Conclusions

In this study, we investigated for the first time the effects of two SNPs in the *TXK* and *PLCE1* genes on growth traits in Ashidan yaks. The results showed that the mutation g.55,999,531C>T in the *TXK* gene affected the body length and chest circumference of yaks, and the mutation g.342,350T>G in the *PLCE1* gene could improve the body height of Ashtin yaks. Therefore, these two SNP loci can be used as potential molecular markers to provide theoretical reference for the breeding and improvement of growth traits in Ashidan yaks.

## Figures and Tables

**Figure 1 animals-14-03506-f001:**
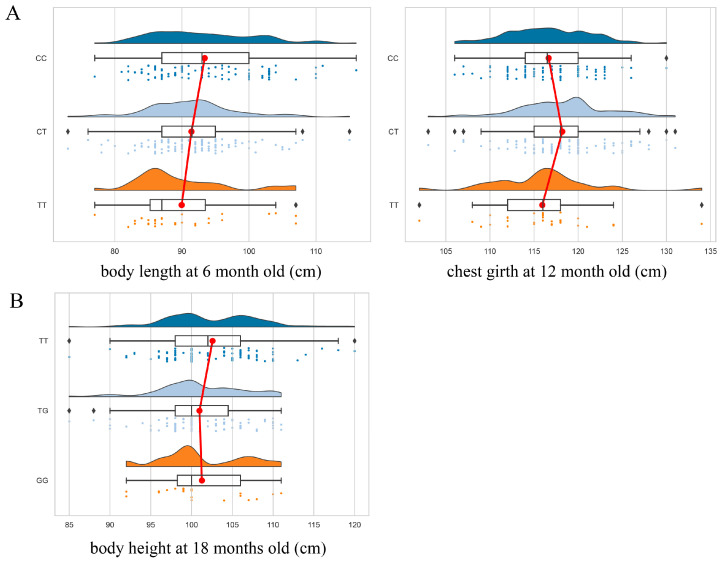
Distribution of all the two genotypes of SNPs. Note: (**A**) *TXK* gene’s g.55,999,531C>T and (**B**) *PLCE1* gene’s g.342,350T>G.

**Table 1 animals-14-03506-t001:** *TXK* and *PLCE1* site amplification primer sequence.

SNPs	Primer Sequence (5′-3′)	Product Size
g.55,999,531C>T	F: GGAAGTGTTGATGTGGTTGGAT	445 bp
	R: TGAAGAGAGCCAGGGAGAAAC	
g.342,350T>G	F: CACCTCCCCAACTCTTCCTA	380 bp
	R: GGCTTCTGTCTATGGGGTCA	

**Table 2 animals-14-03506-t002:** Population genetic information of SNPs loci of the *TXK* and *PLCE1* genes in Ashidan yaks.

SNPs	Genotypic Frequencies			Allelic Frequencies		He	Ne	P	PIC
g.55,999,531C>T	CC	CT	TT	C	T	0.474	1.905	0.102	0.475
	0.397	0.474	0.129	0.634	0.366				
g.342,350T>G	TT	TG	GG	T	G	0.409	1.693	0.366	0.426
	0.487	0.409	0.104	0.692	0.308				

Note: He denotes heterozygosity, Ne signifies the effective number of alleles, and PIC stands for polymorphism information content. A PIC value below 0.25 indicates low polymorphism, between 0.25 and 0.5 signifies moderate polymorphism, and above 0.5 indicates high polymorphism. The *p*-values are reported based on chi-square (X^2^) tests, and a *p*-value less than 0.05 indicates a deviation from Hardy–Weinberg equilibrium.

**Table 3 animals-14-03506-t003:** Association analysis of the *TXK* gene’s g.55,999,531C>T locus with Ashidan yak (*Bos grunniens*) growth traits.

Month of Age	Growth Traits	Genotype (Mean ± SD)			*p*-Value
		CC (92)	CT (110)	TT (30)	
6	BW (kg)	83.69 ± 10.09	86.10 ± 10.32	81.60 ± 11.46	0.07
	WH (cm)	94.16 ± 5.15	94.79 ± 5.49	93.60 ± 6.46	0.51
	BL (cm)	93.40 ± 7.94 ^b^	91.43 ± 6.78 ^ab^	89.97 ± 7.43 ^a^	0.04 *
	CG (cm)	123.51 ± 6.87	124.38 ± 8.73	123.30 ± 8.72	0.69
12	BW (kg)	81.74 ± 12.07	82.85 ± 11.53	82.59 ± 12.44	0.77
	WH (cm)	90.41 ± 3.85	90.88 ± 4.51	89.93 ± 4.38	0.51
	BL (cm)	95.62 ± 4.37	96.19 ± 6.01	95.83 ± 4.33	0.75
	CG (cm)	116.66 ± 4.71 ^ab^	118.20 ± 5.10 ^b^	115.93 ± 5.80 ^a^	0.03 *
18	BW (kg)	121.39 ± 12.00	120.50 ± 12.93	121.04 ± 13.25	0.91
	WH (cm)	99.87 ± 5.41	100.01 ± 4.81	100.77 ± 4.89	0.76
	BL (cm)	101.59 ± 5.61	101.81 ± 5.45	102.36 ± 6.92	0.83
	CG (cm)	137.61 ± 8.62	138.32 ± 10.88	138.09 ± 12.35	0.73
30	BW (kg)	155.51 ± 15.05	152.87 ± 16.16	153.35 ± 18.36	0.60
	WH (cm)	102.23 ± 6.03	102.17 ± 5.07	101.84 ± 6.18	0.95
	BL (cm)	112.31 ± 5.64	113.21 ± 6.06	111.91 ± 6.14	0.54
	CG (cm)	146.44 ± 7.43	147.38 ± 8.41	148.90 ± 7.81	0.43

Note: In the same data group, different lowercase letters indicate significant differences (*p* < 0.05). Abbreviations: BW—body weight; WH—wither height; BL—body oblique length; CG—chest girth. * Indicates a *p* < 0.05, same below.

**Table 4 animals-14-03506-t004:** Association analysis of the *PLCE1* gene’s g.342,350T>G locus with Ashidan yak (*Bos grunniens*) growth traits.

Month of Age	Growth Traits	Genotype (Mean ± SD)			*p*-Value
		TT (113)	TG (95)	GG (24)	
6	BW (kg)	84.75 ± 10.91	84.29 ± 9.90	85.00 ± 10.89	0.93
	WH (cm)	94.31 ± 0.52	94.03 ± 0.56	96.17 ± 1.12	0.23
	BL (cm)	91.73 ± 0.70	92.00 ± 0.76	93.46 ± 1.52	0.59
	CG (cm)	123.97 ± 8.44	123.73 ± 8.28	124.17 ± 6.61	0.96
12	BW (kg)	81.66 ± 13.43	81.37 ± 13.63	82.25 ± 11.07	0.95
	WH (cm)	90.62 ± 4.22	90.49 ± 4.44	90.63 ± 3.60	0.97
	BL (cm)	96.36 ± 2.73	95.55 ± 5.10	95.25 ± 5.99	0.44
	CG (cm)	116.96 ± 5.35	117.51 ± 4.83	117.92 ± 5.03	0.61
18	BW (kg)	107.04 ± 22.89	117.93 ± 22.67	124.35 ± 15.32	0.41
	WH (cm)	101.97 ± 5.27b	101.73 ± 6.16b	104.68 ± 4.12a	0.03 *
	BL (cm)	102.58 ± 5.91	100.98 ± 5.39	101.27 ± 5.46	0.14
	CG (cm)	137.52 ± 9.90	137.77 ± 9.85	141.45 ± 12.17	0.25
30	BW (kg)	153.27 ± 23.83	152.96 ± 15.59	152.88 ± 16.07	0.99
	WH (cm)	100.00 ± 5.38	99.81 ± 4.99	101.24 ± 3.54	0.59
	BL (cm)	113.13 ± 5.83	112.20 ± 6.32	112.05 ± 4.26	0.58
	CG (cm)	146.67 ± 18.09	145.87 ± 7.88	145.82 ± 7.58	0.93

## Data Availability

The data presented in this study are available on request from the corresponding author.

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
