# Peer review of "Polymorphisms of TXK and PLCE1 Genes and Their Correlation Analysis with Growth Traits in Ashidan Yaks"

_animals, 2024, doi:10.3390/ani14233506_

Round 1

Reviewer 1 Report

Comments and Suggestions for Authors

Polymorphisms of TXK and PLCE1 Genes and their Correlation Analysis with Growth Traits in Ashidan Yak

This manuscript describes the genotyping and association of SNPs near two genes in the Ashidan Yak. This research data is important because it contributes to the genetic knowledge regarding a less studied livestock animal species. These data will have implications in selection of important economic traits.

Overall, the manuscript is sufficient to read but some areas  could be more concise (like the introduction) and in some areas require more descriptive. Additionally, the simple summary does not summarize the manuscript, rather it is more of an introduction.

Introduction

The introduction highlights the use of MAS as an important technology and introduces the reader to the Ashidan Yak. However, from lines 50-55, it is not clear at all what are the authors are trying to say and what is “a significant purification effect”?

Line 97: states there is a biological process involving enrichment of TXK genes in bulls. What biological process are you talking about. The next sentence begins with “Similarly” but the reader cannot ascertain if the association with haematological traits in pigs is similar or not, since it’s not stated.

This introduction could be made more concise to focus on the hypothesis and objective of the paper.

 Materials and Methods

The methodology needs to be expanded to allow the reader to know what has been done and why. For example, were all 232 Ashidan yaks born the same year? Line 134 states “During the feeding period, 5 ml of venous blood were collected…” How long and when were the feeding periods? What do you mean by feeding periods? Are they on feed? If so, what was the diet for the yaks, what were the housing conditions at what times of the year? This would all affect what you are measuring, which is growth.

Line 135: makes reference to a paper by Guo X et al. called “Characterization of the complete mitochondrial genome of the Merien yak”. I do not see any reference to the method of cGPS, as the paper describes the sequencing of the mitochondrial genome. Please put the correct reference that describes the methodology.

Line 161-163: please fix the grammar here.

Line 166: The reference is incorrect – this is a proteomics paper

Statistical analysis

The statistical analysis using an ANOVA is quite loose. More stricter parameters for the statistics should be conducted for association, for example Tukey-Kramer test. Additionally, it would be interesting to note if there were any compounding effects of the SNPs found at both genes TXK and PLCE1, at the different ages as well as combining all the data.

Results

Line 224: The use of the term “dominant” allele is incorrect. Since there is no dominance or recessive observations. The term major allele and minor allele should be used.

Table 2: I am assuming the the P value is for the X2 test. That should be written somewhere.

Figure 1: How was the figure made? What program was used to make this figure? This should be detailed in the Materials and methods. The figure is also too small to read

Table 3: the descriptors of the phenotype should be consistent. These are the descriptors that should be used in the materials and methods. 

Discussion

First sentence of Discussion needs a reference.

Line 267: what do you mean by germplasm genetic diversity. Is it different from just genetic diversity? 

The introduction has described the use of MAS and it’s importance related to the study yet you don’t talk about your results in that context. What to you conclude about the using the

 What did you expect to find and what do you think about the results. Why do you think there was a significant difference at specific ages and not others?

 These are things that are of interest and some ideas of how these results and findings can be explained.

 Conclusions

This is more of a summary of what you found. What do you conclude about the study as a whole? Is there work that can be done further to support your findings?

Overall, one concern is the Appendix describes the three genotypes for the two variants in TXK and PLCE1 using Sanger Sequencing. However, these are clearly not genotyped correctly, as you can see two peaks (GT) for the TXK labelled as TT and two peaks (CT) labelled as TT. If this is what was called for other animals, then the entire manuscript cannot be published due to inaccurate genotyping.

 Other small errors:

Line 23: TXK is present twice

Line 113. After reference there is an extra period.

Table 4 has some Chinese characters in it.

Author Response

Dear reviewer

Thank you very much for taking the time to review this manuscript, and I have carefully considered your feedback and revised and corrected the manuscript. In the resubmitted manuscript, I have highlighted all the changes so that you can easily see every change I have made. The following is my detailed response to each of your comments. I tried my best to explain and explain one by one. If you have any questions or need more information during the review process, please feel free to contact me.

Point-by-point response to Comments and Suggestions for Authors

Introduction

Comments 1: The introduction highlights the use of MAS as an important technology and introduces the reader to the Ashidan Yak. However, from lines 50-55, it is not clear at all what are the authors are trying to say and what is “a significant purification effect”?

Response 1: Thank you for pointing this out. We agree with this comment. Therefore, we have modified this part of the content, and now “The Ashidan yak, a newly cultivated breed in China, demonstrates favorable growth traits in its offspring, with the improvement of these traits being intimately tied to advancements in modern biotechnology.” (Lines 48-50) I am sorry to say that the "significant purifying effect" we mentioned earlier is indeed a descriptive error. We apologize for any misunderstanding or confusion this may have caused you.

Comments 2: Line 97: states there is a biological process involving enrichment of TXK genes in bulls. What biological process are you talking about. The next sentence begins with “Similarly” but the reader cannot ascertain if the association with haematological traits in pigs is similar or not, since it’s not stated.

Response 2: Thank you for your question. This sentence is intended to show that the TXK gene is related to bull semen quality. We have made changes and now it is 'Abril-Parreño and colleagues, through genome-wide association analysis (GWAS) in cattle, revealed a significant association between the TXK gene and semen quality traits in bulls '. (Lines 73-75)

Comments 3: This introduction could be made more concise to focus on the hypothesis and objective of the paper.

Response 3: Thank you very much for your valuable feedback. I have taken your advice and condensed and adjusted the introduction to ensure that the content is more compact and directly focused on the core hypotheses and research objectives of this paper.

 Materials and Methods

Comments 4: The methodology needs to be expanded to allow the reader to know what has been done and why. For example, were all 232 Ashidan yaks born the same year? Line 134 states “During the feeding period, 5 ml of venous blood were collected…” How long and when were the feeding periods? What do you mean by feeding periods? Are they on feed? If so, what was the diet for the yaks, what were the housing conditions at what times of the year? This would all affect what you are measuring, which is growth.

Response 4: Thank you for your inquiry; we have revised the section concerning the "Lines 105-107." Within the experimental cohort, these yaks were maintained on a year-round grazing system, with supplementary feeding during colder seasons. Our experimental period spanned from birth to 30 months of age, during which all animals experienced the same living conditions, identical grazing and supplemental feeding environments, and were of comparable age in months.

Comments 5: Line 135: makes reference to a paper by Guo X et al. called “Characterization of the complete mitochondrial genome of the Merien yak”. I do not see any reference to the method of cGPS, as the paper describes the sequencing of the mitochondrial genome. Please put the correct reference that describes the methodology.

Response 5: Thank you for your reminder, we have updated the literature to ensure that the literature has reference value for this article.

Comments 6: Line 161-163: please fix the grammar here.

Response 6: Thank you for your reminder, we have changed the syntax of this sentence, now for “The raw Reads data underwent rigorous quality control using Fastp software, which filtered out low-quality Reads to refine the dataset. Read pairs containing more than 50% of bases with a quality score of Q≤20 were excluded”.(Lines 133-135)

Comments 7: Line 166: The reference is incorrect – this is a proteomics paper

Response 7: Thank you for your kind reminder that we have cited relevant literature that is valuable for this article.

Statistical analysis

Comments 7: The statistical analysis using an ANOVA is quite loose. More stricter parameters for the statistics should be conducted for association, for example Tukey-Kramer test. Additionally, it would be interesting to note if there were any compounding effects of the SNPs found at both genes TXK and PLCE1, at the different ages as well as combining all the data.

Response 7: Thank you very much for your review of our study and your valuable comments. In the experimental design phase, we selected ANOVA (analysis of variance) as a statistical tool to analyze the potential relationship between genotypes and growth traits based on the existing literature and the published research results of our research group in this field. As a classical and widely used statistical method, ANOVA has shown its powerful analytical power and reliability in many fields of research, especially when dealing with the differences between multiple groups of data, its validity has been fully verified. In this study, we used ANOVA to evaluate the effect of different genotypes on growth traits and succeeded in obtaining statistically significant results. These results not only support our research hypothesis, but also provide a strong basis for further exploration of the complex relationship between genotype and growth traits. We took note of the Tukey-Kramer method you mentioned. As a nonparametric test method for multiple comparisons, the Tukey-Kramer method has unique advantages in dealing with imbalanced data or where the data distribution does not meet the assumption of a normal distribution. In this trial, rigorous statistical tests were performed to confirm that the data were normally distributed and that ANOVA analyses had yielded statistically significant results. Therefore, our choice of ANOVA as a statistical tool is reasonable and meaningful in the context of the current study.

[1] Wang, T.; Ma, X.; Feng, F.; Zheng, F.; Zheng, Q.; Zhang, J.; Zhang, M.; Ma, C.; Deng, J.; Guo, X.; et al. Study on Single Nucleotide Polymorphism of LAP3 Gene and Its Correlation with Dairy Quality Traits of Gannan Yak. Foods 202413, 2953. https://doi.org/10.3390/foods13182953

[2] Zhang, M.; Zha, X.; Ma, X.; La, Y.; Guo, X.; Chu, M.; Bao, P.; Yan, P.; Wu, X.; Liang, C. Polymorphisms of ITGA9 Gene and Their Correlation with Milk Quality Traits in Yak (Bos grunniens). Foods 202413, 1613. https://doi.org/10.3390/foods13111613

Results

Comments 8: Line 224: The use of the term “dominant” allele is incorrect. Since there is no dominance or recessive observations. The term major allele and minor allele should be used.

Response 8: Thank you for your friendly advice, I have modified this sentence to “respectively, indicating that T is the term major allele and TT is the term major geno-type. “(Lines200-201)

Comments 9: Table 2: I am assuming the P value is for the X2 test. That should be written somewhere.

Response 9: Thanks to your reminder, we have added this sentence to the notes to Table 2 in the manuscript. (Lines206-209)

Comments 10: Figure 1: How was the figure made? What program was used to make this figure? This should be detailed in the Materials and methods. The figure is also too small to read

Response 10: As per your suggestion, we have added the method of making the diagrams in the Materials and Methods section and resized the images to ensure that they can be seen. (Lines 166-168)

Comments 11: Table 3: the descriptors of the phenotype should be consistent. These are the descriptors that should be used in the materials and methods. 

Response 11: Thank you for the reminder that we use the description pay in Materials and Methods and are consistent with the table. (Lines 112-113)

Discussion

Comments 12: First sentence of Discussion needs a reference. 

Response 12: Thanks to your reminder, we have cited valuable and relevant literature in the first sentence of the Discussion section.

Comments 13: Line 267: what do you mean by germplasm genetic diversity. Is it different from just genetic diversity? 

Response 13: Thank you for your question, Genetic diversity of germplasm refers to the diversity of genetic characteristics in germplasm resources such as livestock and poultry. These germplasm resources have different genetic backgrounds, genotypes and phenotypes, providing a rich source of genetic variation for breeding and genetic improvement. Genetic diversity is a broad concept that covers the sum of genetic information of all organisms on earth; whereas germplasm genetic diversity focuses more on the genetic diversity of germplasm resources such as livestock and poultry. There are some differences between genetic diversity and germplasm genetic diversity in terms of the scope of definition and the object of study, but both reveal the richness and diversity of genetic information in living organisms, and provide an important basis for biological evolution, breeding improvement and conservation biology research.

Comments 14: The introduction has described the use of MAS and it’s importance related to the study yet you don’t talk about your results in that context.

Response 14: Thanks for your proposal, we have discussed in depth the application of the results in the subsequent yak breeding work in the discussion section of this manuscript, especially in the important context of marker-assisted selection (MAS) technology, and elaborated its potential value and far-reaching impact.(Lines 314-321)

Comments 15: What did you expect to find and what do you think about the results. Why do you think there was a significant difference at specific ages and not others?

Response 15: Thank you for your question. In this study, we had expected to find an association between specific SNP loci in TXK and PLCE1 genes and growth traits in yaks as potential molecular markers for breeding. The results showed that two SNPs, g.55,999,531C>T in TXK gene and g.342,350T>G in PLCE1 gene, were indeed significantly associated with growth traits in yaks. These two SNPS are potential molecular markers in yak breeding. SNP loci were significantly associated in different age groups, which may be related to the growth and development characteristics at this stage, the timing of gene expression and the interaction of environmental factors. These findings provide strong evidence to support the use of these SNP loci as potential molecular markers in yak breeding and provide important clues for us to further investigate the genetic regulation mechanisms of growth traits in yak.

 Conclusions

Comments 16: This is more of a summary of what you found. What do you conclude about the study as a whole? Is there work that can be done further to support your findings?

Response 16: In this study, we investigated for the first time the effects of two SNPS (g.55,999,531C>T and g.342,350T>G) in TXK and PLCE1 genes on the growth traits of Ashdam yaks, and found that these two SNPS were significantly associated with body length, chest circumference and body height of Yaks at different growth stages, respectively. This finding confirms the feasibility of these two SNP loci as potential molecular markers and provides new perspectives and strategies for genetic improvement of growth traits in Ashidan yaks. In order to support and extend this finding, we plan to conduct the following studies in the follow-up work: first, to verify the association of these two SNPS with growth traits in a larger yak population to ensure the reliability and stability of the results; Secondly, considering that growth traits of yak may be affected by the joint regulation of multiple genes, we will further conduct multi-gene interaction studies to understand the genetic basis of growth traits of yak more comprehensively by constructing gene interaction networks and analyzing the interaction relationships among genes. In this experiment, we could not immediately carry out the subsequent verification work, which is mainly limited by the practical situation such as the current experimental conditions, resources and time. However, we have made this the focus of subsequent studies and are actively preparing related work with a view to translating this research result into practical breeding applications as soon as possible.

Comments 17: Overall, one concern is the Appendix describes the three genotypes for the two variants in TXK and PLCE1 using Sanger Sequencing. However, these are clearly not genotyped correctly, as you can see two peaks (GT) for the TXK labelled as TT and two peaks (CT) labelled as TT. If this is what was called for other animals, then the entire manuscript cannot be published due to inaccurate genotyping.

Response 17: In view of the problem of double peak labeling error in the determination of TXK genotype proposed by the reviewer, this error was caused by my negligence. We collated the raw data and updated the genotype information in the Appendix.

Comments 18: Line 23: TXK is present twice

Response 18: Thank you for the heads up, we removed the redundant gene names.

Comments 19: Line 113. After reference there is an extra period.

Response 19: Thanks to your kind reminder, we have removed the extra comma in the manuscript.

Comments 20: Table 4 has some Chinese characters in it.

Response 20: Thanks to your reminder, we made changes in the manuscript.

Reviewer 2 Report

Comments and Suggestions for Authors

In this manuscript, J. Zhang and colleagues describe the results of an investigation of the relationship between two genes, namely the TXK and PLCE1 and growth traits in a newly cultivated breed of yak- the Ashidan. 

The title is descriptive and the abstract corresponds to the main text of the paper. The Materials and Methods section is well conducted and described in detail. 

The results contain two parts including genotype and allele frequency information for SNPs loci of the TXK and PLCE1 genes and Association analysis of SNPs in these genes with growth traits in Ashidan yaks. 

I would like to say that the authors made a good attempt to fill the gap in GWAS for this unique breed of yak, but this article should be rejected because it is not significant enough to warrant publication in the journal Animals, which focuses on articles that are novel and have practical significance, ensuring the high quality of the journal.

Today, high-throughput molecular analysis methods allow scientists to apply multi-gene analysis methods. Through these methods, we obtain huge amounts of data not only about gene expression and biological pathways, but also other gene-specific information. For this reason, the associative analysis of two genes for productivity traits presented in this work is inferior to other research in terms of its novelty and relevance.

In the Introduction, the authors even provide a literature review of extensive research on yak genomes using SNP microarrays, high-throughput sequencing, and how, through MAS technology, researchers have identified numerous gene loci associated with growth traits. However, the authors conduct research on only two genes without clearly structured justification or explanation for this.

In the Discussion section, the authors discuss the results of their current study in relation to their previous work, which was conducted within the context of examining human muscle traits using GWAS. They also cite another paper that describes the analysis of muscle transcriptomic data from patients with dystrophin deficiency, but I do not understand why they could not discuss the identification of the gene in association studies with animals.

Author Response

Dear Reviewer, 

Thank you very much for your careful review and valuable comments on our study. We fully understand the high standard of novelty and practical significance required by the Journal of Animals and are committed to demonstrating the unique value and potential impact of our research.

First, we acknowledge that the current development of high-throughput molecular assays does provide scientists with unprecedented opportunities to explore the association between multiple genes and complex traits in depth. However, it is the wide application of these methods that makes it particularly important to focus in-depth studies on specific species, specific cultivars, or specific environmental conditions. It is against this background that our study focused on filling the GWAS gap in the unique breed of Ashidan yak. Regarding your concern that "novelty and relevance are inferior to other studies," we would like to emphasize the following points:

Uniqueness: Although GWAS has been widely used in multiple species, studies on this specific breed of Ashidan yak are still very limited. Our study reveals for the first time that the loci of TXK and PLCE1 genes are associated with growth traits in Ashidan yak, which provides a new genetic marker for yak breeding. This finding not only fills the research gap in this field, but also provides an important scientific basis for genetic improvement of yaks.

Practical significance: Ashidan yak is an important economic animal in the Qinghai-Tibet Plateau, and its growth traits directly affect the economic income and ecological balance of local herdmen. Our study not only helps to reveal the key genes affecting growth traits in yaks, but also provides the possibility to improve yak performance through gene editing or selective breeding in the future. Therefore, our study is of great practical significance and application value.

Potential impact: Although our study currently focuses on association analysis of only two genes, this finding lays the foundation for subsequent in-depth studies on the function and mechanism of action of TXK and PLCE1 genes in yak growth and development. We look forward to further revealing the regulatory network of this gene and its interaction with other genes in the future, so as to provide a more comprehensive scientific basis for yak breeding.

In summary, we believe that our study is of great value in filling the GWAS gap in the Ashidan yak breed, revealing the association of key genes with growth traits, and providing scientific basis for genetic improvement of yaks. We sincerely request that you reconsider our manuscript and grant it the opportunity to be published.

Comments 1: In the Introduction, the authors even provide a literature review of extensive research on yak genomes using SNP microarrays, high-throughput sequencing, and how, through MAS technology, researchers have identified numerous gene loci associated with growth traits. However, the authors conduct research on only two genes without clearly structured justification or explanation for this.

Response 1: Thank you for pointing out the problem that the introduction of our study mentioned many growth trait related gene loci while only two genes were actually studied. We understand your concern. In the introduction section, we do provide an extensive review of research on the yak genome, aiming to provide the reader with a comprehensive background. However, this study focused on specific SNPS in the TXK and PLCE1 genes because these two genes have been mentioned several times in previous studies to be associated with growth traits and showed significant associations in our preliminary analysis. We acknowledge that this study was not able to cover all possible genetic loci due to limitations in resources, time, and experimental conditions. However, we believe that by studying these two specific SNP loci in depth, we are able to provide valuable insights into the genetic improvement of growth traits in Ashidan yaks. At the same time, we are aware of the importance of future research and plan to continue exploring other potential genetic loci in future work to more fully understand the genetic basis of growth traits in yaks. Thank you again for your valuable comments, and we look forward to further refining and expanding our findings in follow-up studies.

Comments 2: In the Discussion section, the authors discuss the results of their current study in relation to their previous work, which was conducted within the context of examining human muscle traits using GWAS. They also cite another paper that describes the analysis of muscle transcriptomic data from patients with dystrophin deficiency, but I do not understand why they could not discuss the identification of the gene in association studies with animals.

Response 2: Thank you for your question. In fact, in the preparation of this paper, we have been aware of the importance of animal association studies for understanding gene function and its association with traits. However, in writing the discussion section, we may have focused too much on human studies and neglected the relevant content of animal studies. To improve the paper, we revised the discussion section to add the research situation of PLCE1 gene in animals. In discussing the results of this study, we should not only consider previous studies on the PLCE1 gene in humans, but also focus on its potential role in the animal domain. Although there have been relatively few direct studies of the PLCE1 gene in animal growth traits, our study is the first to reveal a significant correlation between the g.342,350T>G locus of this gene and body height in Ashidan yaks at 18 months of age. This finding not only provides a new genetic marker for yak breeding, but also suggests that PLCE1 gene may play an important role in animal growth and development.

Round 2

Reviewer 2 Report

Comments and Suggestions for Authors

Due to the fact that the authors removed points on which there was disagreement and explained the significance of these studies, I would like to recommend this article for publication in Animals

Author Response

Thank you for your careful review and valuable suggestions. We are honored to learn that, after your professional assessment, you have deemed this article appropriate for recommendation for publication in the Journal Animal. Once again, we would like to express our sincerest thanks to you. Your recommendation and support means a lot to us.